# Inter-arm difference in systolic blood pressure: Prevalence and associated factors in an African population

**Gwladys Nadia Gbaguidi**[1,2]*, **Audrey Kaboure**[2], **Yessito Corine Houehanou**[1,3], **Salimanou Ariyo Amidou**[1], **Dismand Stephan Houinato**[1,2,4], **Victor Aboyans**[2,5], **Philippe Lacroix**[2,6]

1 Faculty of Health Sciences, Laboratory of Chronic and Neurologic Diseases Epidemiology, LEMACEN, University of Abomey-Calavi, Cotonou, Benin, 2 INSERM, Univ. Limoges, CHU Limoges, IRD, U1094 Tropical Neuroepidemiology, Institute of Epidemiology and Tropical Neurology, GEIST, Limoges, France, 3 ENATSE, University of Parakou, Parakou, Benin, 4 Neurology Unit, CNHU Cotonou, Cotonou, Benin, 5 Department of Cardiology, CHU Limoges, Limoges, France, 6 Department of Vascular Medicine, CHU Limoges, Limoges, France

* nadiagbaguidi@gmail.com

**Data Availability Statement:** Data cannot be shared publicly because of confidentiality. Data are available from the UMR Inserm 1094 NET Institutional Data Access (contact via 2 rue du Dr

## Abstract

### Objectives

Inter-arm blood pressure difference (IABPD) can lead to underdiagnosis and poor management of hypertension, when not recognized and are associated with increased cardiovascular mortality and morbidity. However, the prevalence and associated risk factors of IABPD in sub-Saharan Africa are unknown. This study aims to determine the prevalence and associated risk factors of IABPD among Tanve Health Study (TAHES) participants, a cohort about cardiovascular diseases in a rural area in Benin.

### Methods

The cohort was conducted since 2015 among adults aged 25 years and over in Tanve village. Data were collected from February to March, 2020. Brachial blood pressure were recorded at rest on both arm with an electronic device. Systolic IABPD (sIABPD) was defined as the absolute value of the difference in systolic blood pressure between left and right arms $\geq$ 10 mmHg. A multivariate logistic regression models identified factors associated with sIABPD.

### Results

A total of 1,505 participants (women 59%) were included. The mean age was 45.08 ±15.65 years. The prevalence of sIABPD $\geq$ 10 mmHg was 19% (95%CI: 17–21). It was 19% (95% CI: 16–22) in men and 20% (95%CI: 17–22) in women. In final multivariable model, the probability of sIABPD $\geq$ 10 mmHg increased significantly with age (adjusted OR (aOR) = 1.1; 95%CI: 1.02–1.20 per 10-years), hypertension (aOR = 2.33; 95%CI: 1.77–3.07) and diabetes (aOR = 1.96; 95%CI: 1.09–3.53).

Marcland, 87025 LIMOGES Cedex - Tél.: 05 55 43
58 20) for researchers who meet the criteria for
access to confidential data. To access the data, a
request can be sent to: pierre-marie.preux@unilim.
fr.

**Funding:** The Foundation for high blood pressure
research, Paris, France, supported this survey. The
sponsors had no role in the design, methods,
participant's recruitment, data collection, analysis,
or preparation of this manuscript.

**Competing interests:** The authors have declared
that no competing interests exist.

## Conclusion

Almost quarter of sample have a sIABPD $\geq$ 10 mmHg, with an increased risk with older age
and hypertension and diabetes.

## Introduction

Cardiovascular diseases (CVD) is the leading cause of death worldwide [1]. Hypertension is
one of the most important modifiable cardiovascular risk factors (CVRFs) and can be con-
trolled by lifestyle changes or drug therapy, hence the interest in blood pressure (BP) measure-
ments in high-risk populations [2]. BP measurement is part of routine clinical examination,
especially as the detection of hypertension is a key component of clinical cardiovascular assess-
ment [3]. Bilateral measurement of BP in both arms is recommended by many guidelines at
initial visit and then annually [4, 5], for preventing misdiagnosis of hypertension [6]. The risk
of cardiovascular morbidity and mortality seems to increase with inter-arm blood pressure dif-
ference (IABPD) [7, 8]. Indeed, although a threshold of IABPD of 10 mmHg is admitted in
clinical practice, any difference beyond 5 mmHg is proportionally associated with cardiovas-
cular and mortality risk increase [8–13].

The vast majority of the studies on IABPD were performed in the developed countries [14–
18]. Data on the prevalence and associated risk factors of IABPD in sub-Saharan Africa are
scarce. As CVRFs are increasing in low- and middle-income countries, the assessment of
IABPD, an easy and inexpensive risk predictor is of high interest in this setting. Taking the
opportunity of a population-based study in Benin, we aimed to determine the prevalence and
associated risk factors of IABPD in a rural community of Benin.

## Methods

### Design and population study

Our study is based on data from the TAnve Health Study (TAHES) cohort. This is a prospective
cohort in Benin since February 2015, in the two neighbouring villages of Tanve and Dékanme,
located in the commune of Agbangninzoun, 150 km away from Cotonou, the economic capital of
Benin (S1 File). Tanvè has health center including a dispensary and a maternity [19]. This cohort
includes people resident at least 6 months in the villages of Tanvè or Dékanmè, aged 25 years and
over. Participants' consents were obtained. Pregnant woman and participants unable to answer
the questions were excluded from the study. This cross-sectional study on IABPD used the fifth
annual survey of TAHES conducted from February 8 to March 1, 2020.

### Data collection

Data was collected during a systematic door-to-door survey by 7 team of 2 trained investiga-
tors, according to the WHO STEPS methodology [20]. Demographic, lifestyle (alcohol,
tobacco, sedentary, fruit and vegetable consumption), history of diseases (hypertension, diabe-
tes), weight, height, brachial BP, blood glucose and proteinuria data were collected using a
questionnaire adapted from WHO STEPS tools.

### Blood pressure measurements

Brachial BP were recorded in both arms, using an electronic device (OMRON M3, HEM-
7131), with adequate cuffs for normal and large arms. Three measures were performed on

both arms at three minutes intervals, in seated position after a rest of at least 15 minutes. On each arm, BP was defined as the average of the last two measurements. sIABPD ≥10 mmHg was defined as the absolute value of the difference for SBP between the left and right arms greater than or equal to 10 mmHg [9].

## Other variables

Covariates were defined according to WHO recommendations for STEPS surveys [20]. Low fruit and vegetable consumption was defined as less than five portions (400 grams) of fruits and vegetables per day. Current and former (less than one year) smokers were considered as active smokers. Sedentary lifestyle was considered as sitting/sleeping for more than 8 hours daily, outside the night-sleep period. Body Mass Index (BMI) was calculated as weight in kilograms divided by the square of height in meters. The BMI was categorized in four groups: underweight ($<18.5kg/m^2$), normal (from 18.5 to 24.9 $kg/m^2$), overweight (from 25 and 29.9 $kg/m^2$), and obesity ($\geq 30kg/m^2$). Hypertension was defined as systolic and/or diastolic blood pressure $\geq$ 140/90 mmHg in the highest of the two arms, or whether receiving anti-hypertensive medication. Diabetes was defined by fasting capillary whole blood glucose value $\geq$ 7 mmol/L or currently taking diabetes medication. Semi-quantitative proteinuria was assessed based on urine protein dipstick and defined by the color change of an indicator (from 'trace ' to '++++') [21]. Anxiety and depression were respectively assessed using Goldberg Anxiety Scale (GAS) and Goldberg Depression Scale (GDS) [22]. Each global score, ranges from 0 to 18, and questions/items were based on responses "yes" or "no", rated one or zero point respectively. Anxiety was defined by GAS $\geq$ 5 and depression by GDS $\geq$2 [22]. Data on history of cardiovascular or neurological disease, such as peripheral arterial disease, heart failure, angina, and stroke, were based on previous diagnosis by a professional health care.

## Ethics approval and consent to participate

The TAHES protocol received approval No. 026 of August 28, 2014, from the National Committee of Ethics for Health Research (CNERS) of the Ministry of Health of Benin. Informed and written consent was required for each participant before inclusion in TAHES. Furthermore, a physician recruited for the study examined participants with abnormal BP. Following their examinations, they received counseling, prescription drugs and were referred to the Abomey municipal health center for further exams or the regional hospital for cardiologic consultation, depending on the participant's situation.

## Statistical analysis

All analyses were performed using the software R (version 3.6.2). Baseline characteristics of study participants were described and compared according to sIABPD $\geq$ 10 mmHg status using chi-square or Fisher's tests for qualitative variables and Wilcoxon test for quantitative variables. The prevalence of sIABPD $\geq$ 10 mmHg was estimated. This prevalence has been described according to gender and age and compared using chi-square test. Distribution of sIABPD was performed. The prevalence of sIABPD was also described according to SBP classification defined by ESC 2018 guidelines (optimal <120 mmHg, normal: 120–129 mmHg, High normal: 130–139, Grade 1 Hypertension: 140–159, Grade 2 Hypertension: 160–179, Grade 3 Hypertension $\geq$ 180 mmHg) [5]. Associated factors to sIABPD were identified by using logistic regression models. We have tested association between sIABPD with each covariates and p value less than 0.20 were included in multivariable analysis, along with age and gender systematically.

The interactions between variables included in multivariable model were tested. In addition linearity of quantitative variables on logit of the probability to have a sIABPD $\geq$ 10 mmHg was checked. We proceeded by backward stepwise selection to obtain the final model. In the sensitivity analysis, the factors associated with sIABPD $\geq$ 15 mm Hg were identified using a logistic regression model. Odds ratio (OR) and their confidence intervals (CI) at 95% were reported and a p value less than 0.05 was considered as statistically significant.

## Results

### Study population

A total of 1,571 participants were included in the TAHES cohort in 2020. Among them, 66 pregnant women were excluded. Thus, 1505 participants were included in this study (Fig 1). The mean age was 45.08 ±15.65 years. The sex ratio (male/female) was 0.7 (Table 1).

The mean of Systolic blood pressure (SBP) and diastolic blood pressure (DBP) of participants were 126.9±20.2 and 83.5 ±12.9 mmHg, respectively. In our study population, 518 (34.4%) participants had hypertension; among them 161 (10.7%) were on treatment.

### Prevalence of sIABPD

Of the participants, 292 had a sIABPD $\geq$ 10 mmHg for a prevalence estimated at 19.4% (95% CI: 17.4%-21.5%). Among 1,505 participant's, there were 11.4%, 4.9% and 3.0% with a sIABPD in the ranges: 10-14mmHg, 15-19mmHg and $\geq$ 20mmHg respectively. Frequency of sIABPD $\geq$ 10 mmHg was almost the same in men and in women in all sIABPD groups except for the one $\geq$ 20 mmHg in which sIABPD was slightly more frequent in women (Fig 2). In participants aged 45 to 54 years, the prevalence of sIABPD $\geq$ 10 mmHg was significantly

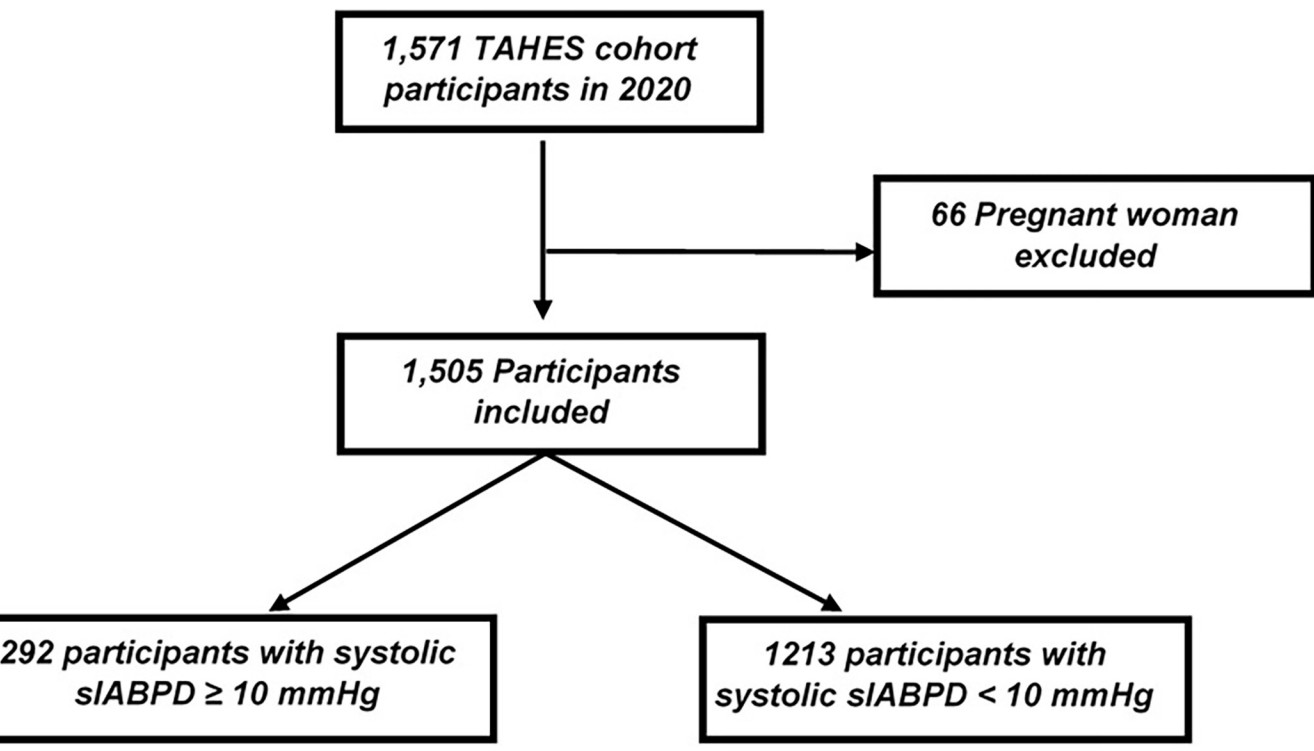

**Fig 1. Flowchart of inclusion in this study population from TAHES cohort, 2020.**

**Table 1.** Systolic inter arm blood pressure difference groups according to socio-demographic characteristics, behavioural characteristics, anxiety, depression and proteinuria, TAHES study, Benin 2020.

| | Total (N = 1505) | sIABPD ≥ 10 mmHg (N = 292) | sIABPD < 10 mmHg (N = 1213) | p-value[Ł] |
|---|---|---|---|---|
| | Mean (sd) / n (%) | Mean (sd) / n (%) | Mean (sd) / n (%) | |
| **Age (years)** | 45.08 ±15.7 | 49.1 ± 16.7 | 44.1±15.2 | <10−5 |
| **Gender** | | | | |
| Female | 893 (59.3) | 176 (60.3) | 717 (59.1) | 0.7663 |
| Male | 612 (40.7) | 116 (39.7) | 496 (40.9) | |
| **Education levels** | | | | |
| Illiterate | 1019 (67.7) | 207 (70.9) | 812 (66.9) | |
| Less than primary level | 256 (17.0) | 43 (14.7) | 213 (17.6) | 0.3976 |
| Primary level and above | 230 (15.3) | 42 (14.4) | 188 (15.5) | |
| **Marital status** | | | | |
| In couple | 1261 (83.8) | 229 (78.4) | 1032 (85.1) | <10−5 |
| Single, widowed or divorced | 244 (16.2) | 63 (21.6) | 181 (14.9) | |
| **Occupation** | | | | |
| Small self-employed without trade register | 907 (60.3) | 167 (57.2) | 740 (61.0) | |
| Independent farmer/contractor | 243 (16.1) | 58 (19.9) | 185 (15.3) | 0.0688 |
| Small business employee/farm worker | 144 (9.6) | 21 (7.2) | 123 (10.1) | |
| Private employee or official worker | 43 (2.9) | 6 (2.1) | 37 (3.1) | |
| Retired/unemployed/other/student/apprentice | 168 (11.2) | 40 (13.7) | 128 (10.6) | |
| **Monthly income ($US)** | | | | |
| < 68 | 832 (55.3) | 155 (53.1) | 677 (55.8) | |
| 68–117 | 380 (25.2) | 77 (26.4) | 303 (25.0) | 0.6997 |
| ≥ 117 | 293 (19.5) | 60 (20.5) | 233 (19.2) | |
| **Tobacco smoking** | 75 (5.0) | 21 (7.2) | 54 (4.5) | 0.0748 |
| **Low intake of fruit & vegetable** | 756 (50.2) | 138 (47.3) | 618 (50.9) | 0.2863 |
| **Sedentarity behaviour** | 162 (10.8) | 37 (12.7) | 125 (10.3) | 0.2864 |
| **Alcohol consumption last 30 days** | 754 (50.1) | 148 (50.7) | 606 (50.0) | 0.8748 |
| **BMI (Kg/m$^2$)** | | | | |
| Normal | 870 (57.8) | 166 (56.8) | 704 (58.0) | |
| Underweight | 218 (14.5) | 46 (15.8) | 172 (14.2) | 0.2394 |
| Overweight | 283 (18.8) | 47 (16.1) | 236 (19.5) | |
| Obesity | 134 (8.9) | 33 (11.3) | 101 (8.3) | |
| **Cardiovascular or neurological history** | | | | |
| Peripheral arterial disease | 7 | 0 | 7 | |
| Heart failure | 4 | 0 | 4 | 0.278 |
| Angina pectoris | 5 | 1 | 4 | |
| Stroke | 4 | 2 | 2 | |
| **Hypertension** | 518 (34.4) | 155 (53.1) | 363 (29.9) | <10−5 |
| **Diabetes** | 55 (3.7) | 19 (6.5) | 36 (3.0) | <10−5 |
| **Anxiety** | 266 (17.7) | 65 (22.3) | 201 (16.6) | <10−5 |
| **Depression** | 555 (36.9) | 121 (41.4) | 434 (35.8) | 0.0833 |
| **Proteinuria** | 56 (3.7) | 8 (2.7) | 48 (4.0) | 0.4153 |

sIABPD: Systolic inter arm blood pressure difference

[Ł]: p value of chi-square or Fisher's tests for qualitative variables and Wilcoxon test for quantitative variable

higher in men than women. In contrast, there was no statistically significant difference in the prevalence of sIABPD ≥ 10 mmHg according to gender in the other age groups (Fig 3). The prevalence of sIABPD increased with the rise in SBP (Fig 4).

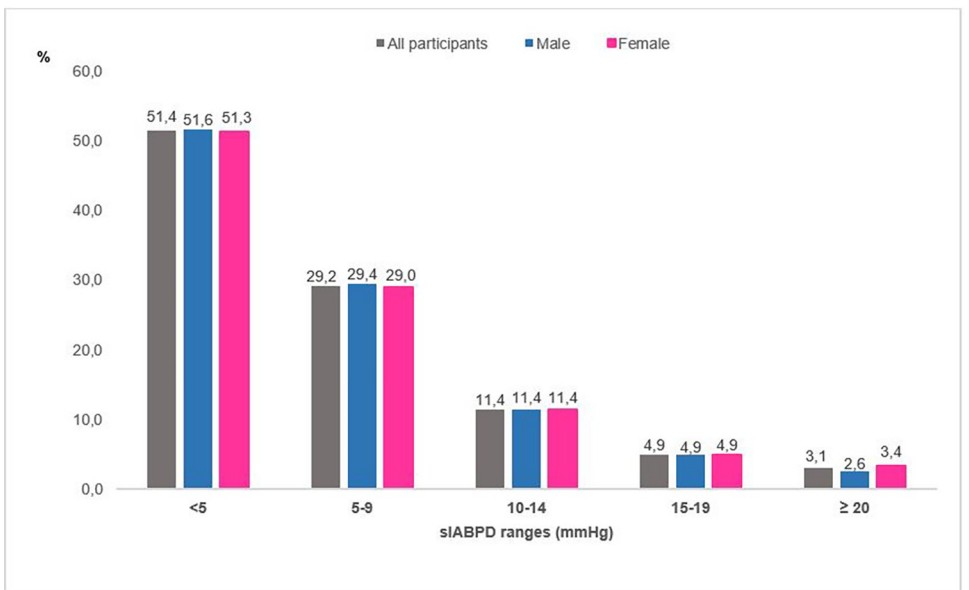

**Fig 2. Distribution of absolute systolic inter-arm blood pressure difference (sIABPD) by gender, TAHES study, Benin 2020.**

## Risk factors for sIABPD

Univariate logistic regression has shown a higher probability of sIABPD ≥ 10mmHg per 10-years of age (p<0.001). Increasing age, living alone, hypertension and diabetes were

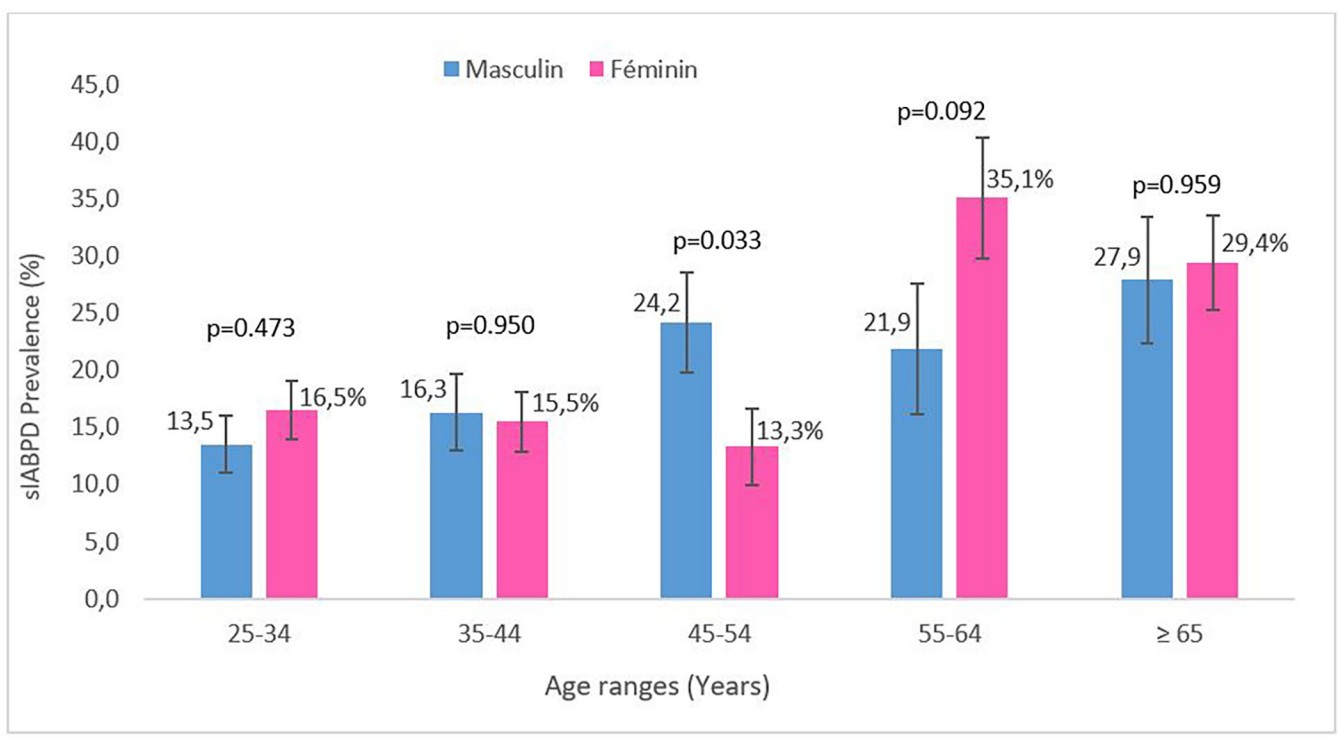

**Fig 3. Gender prevalence of systolic inter-arm blood pressure difference (sIABPD) ≥ 10 mmHg by age groups, TAHES study, Benin 2020.**

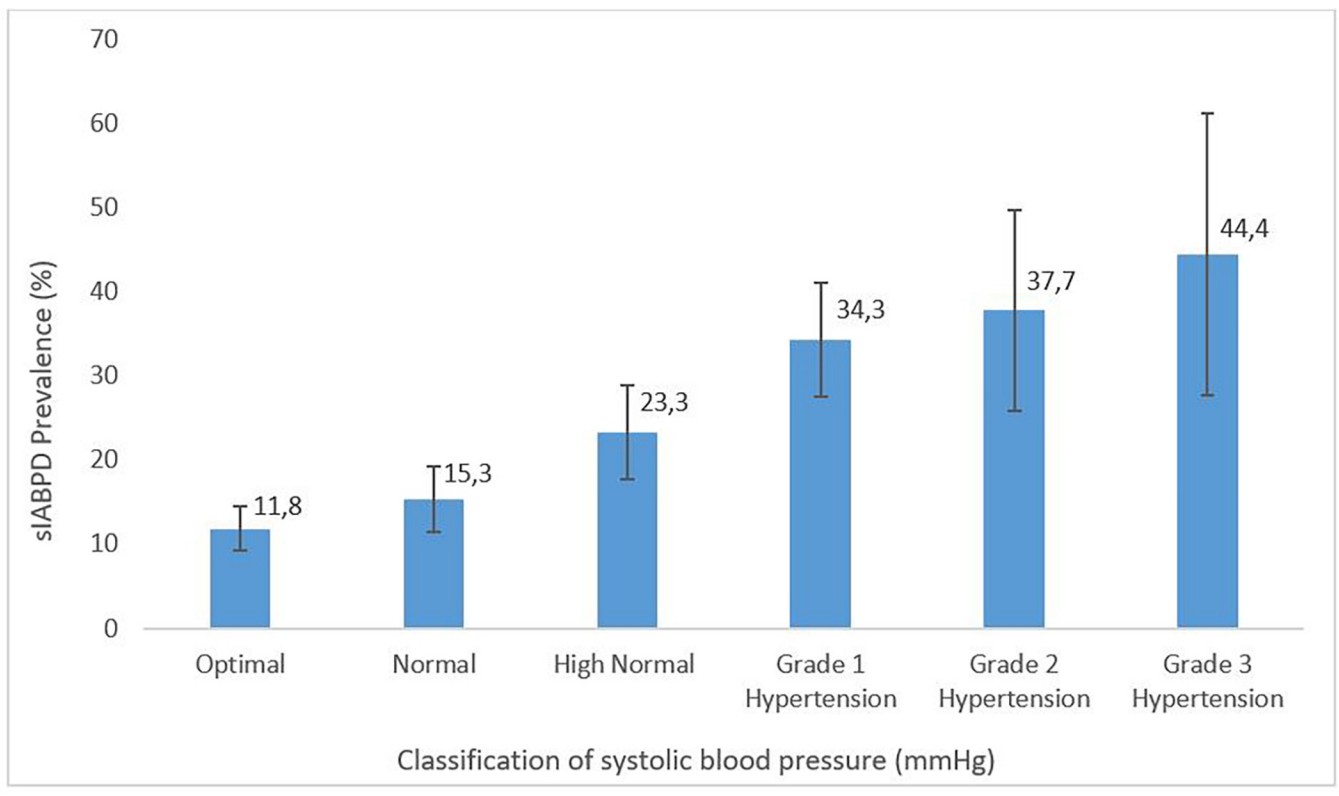

**Fig 4. Prevalence of sIABPD ≥ 10 mmHg by systolic blood pressure classification, TAHES study, Benin 2020.**

associated with higher prevalence of sIABPD ≥ 10 mmHg (Table 2). In a multivariate model adjusted for age and gender, both diabetes and hypertension were independently associated with sIABPD ≥ 10 mmHg (Table 3).

In sensitivity analysis, hypertension and education levels were statistically associated with sIABPD ≥ 15 mmHg (S2 File).

## Discussion

In this study, we found a very high prevalence (19.4%) of sIABPD ≥ 10 mmHg. Older age, hypertension and diabetes was associated to sIABPD ≥ 10 mmHg. To our knowledge, this is the first report from a study in an African population.

The prevalence of sIABPD ≥ 10 mmHg in this study was higher than some results of previous studies conducted in general populations [23–25]. In Japan, Kimuraa et al. reported a prevalence of 9.1% [24]. In Finland, Johansson et al. reported a prevalence of 10.1% in general population [23]. In the Framingham Heart Study, 9.4% of participants had a sIABPD ≥10 mmHg [25]. Our higher prevalence might be explained by differences in methods used for BP measurement and study populations. In our study, we performed sequentially three successive BP measures on each arm in a seated position. Kimuraa et al. in the Ohasama study measured BP simultaneously in both arms only two times but in supine position [24]. Also, in the Framingham Heart Study, BP was measured sequentially three times in supine position [25]. The guidelines for hypertension recommend repeated BP measurement in seated position to acquire accurate BP values [26, 27]. However, according to a meta-analysis, the number of subjects with a systolic and diastolic IABPD ≥10 mm Hg was significantly lower when BP

**Table 2. Factors associated with systolic inter-arm blood pressure difference ≥ 10 mmHg, univariate analysis, TAHES study, Benin 2020.**

| | Univariate analysis | |
|---|---|---|
| | Crude OR [CI 95%] | p-value[Ł] |
| **Age (per 10 years)** | 1.21 [1.12–1.31] | < 0.001* |
| **Gender** | | |
| Female (vs. male) | 1.05 [0.81–1.36] | 0.716 |
| **Education levels** | | |
| Illiterate | 1 | 0.39 |
| Less than primary level | 0.79 [0.55–1.14] | |
| Primary level and above | 0.88 [0.61–1.27] | |
| **Marital status** | | |
| Single, widowed or divorced (vs. in couple) | 1.57 [1.14–2.16] | 0.006* |
| **Occupation** | | |
| Small self-employed without trade register | 1 | 0.07 |
| Independent farmer/contractor | 1.39 [0.99–1.95] | |
| Small business employee/farm worker | 0.76 [0.46–1.24] | |
| Private employee or official worker | 0.72 [0.30–1.73] | |
| Retired/unemployed/other/student/apprentice | 1.38 [0.93–2.05] | |
| **Monthly income ($US)** | | |
| < 68 | 1 | 0.7 |
| 68–117 | 1.11 [0.82–1.51] | |
| ≥ 117 | 1.12 [0.81–1.57] | |
| **Tobacco smoking** (vs non-smokers) | 1.66 [0.99–2.80] | 0.056 |
| **Low intake of fruit & vegetable** (Yes vs No) | 0.86 [0.67–1.11] | 0.258 |
| **Sedentarity behavior** (Yes vs No) | 1.26 [0.85–1.87] | 0.242 |
| **Alcohol consumption last 30 days** *(Yes vs No)* | 1.03 [0.80–1.33] | 0.824 |
| **BMI (Kg/m$^2$)** | | |
| Normal | 1 | 0.249 |
| Underweight | 1.13 [0.79–1.64] | |
| Overweight | 0.84 [0.59–1.21] | |
| Obesity | 1.39 [0.90–2.13] | |
| **Hypertension** (Yes vs No) | 2.65 [2.04–3.44] | < 0.001* |
| **Diabetes** (Yes vs No) | 2.28 [1.29–4.03] | 0.005* |
| **Anxiety** (Yes vs No) | 1.44 [1.05–1.97] | 0.023* |
| **Depression** (Yes vs No) | 1.27 [0.98–1.65] | 0.072 |
| **Proteinuria** (Yes vs No) | 0.68 [0.32–1.46] | 0.327 |

OR: Odd ratio

[Ł:] p value of Wald test for binary variables or likelihood test for categorical variables with more than two modalities

*: Statistically significant

measurements were performed simultaneously instead of sequentially [12]. This could have overestimated the prevalence of sIABPD ≥10 in this study.

Our prevalence was similar with pooled prevalences of the sIABPD ≥ 10 mmHg found by Clark et al. in a systematic review (19.6%) [7]. In the studies retained in their review, most of the participants were patients with high risk of cardiovascular factors/outcomes (hypertensive, diabetics, and patients with renal or vascular disease). The prevalence of cardiovascular risk factors such as hypertension (34.4%), diabetes (3.7%) and anxiety (17.7%) are also high in our study. This could explain the high prevalence of sIABPD observed [7]. In fact depression and

**Table 3. Factors associated with *systolic inter-arm blood pressure difference (sIABPD)* ≥ *10 mmHg*, multivariable analysis, TAHES study, Benin 2020.**

| | Initial model | | Final model[1] | |
|---|---|---|---|---|
| | aOR (95%CI) | p value[Ł] | aOR (95%CI) | p value[Ł] |
| **Age (per 10 years)** | 1.07 [0.97–1.18] | 0.181 | 1.11 [1.02–1.21] | 0.012* |
| **Gender** | | | | |
| Female (vs. male) | 1.01 [0.76–1.34] | 0.959 | 1.03 [0.79–1.35] | 0.83 |
| **Marital status** | | 0.465 | | |
| Single, widowed or divorced (vs. couple) | 1.15 [0.79–1.69] | | | |
| **Occupation** | | 0.423 | | |
| Small self-employed without trade register | 1 | | | |
| Independent farmer/contractor | 1.22 [0.85–1.76] | | | |
| Small business employee/farm worker | 0.76 [0.46–1.25] | | | |
| Private employee or official worker | 0.65 [0.26–1.60] | | | |
| Retired/unemployed/other/student/apprentice | 0.99 [0.64–1.53] | | | |
| **Tobacco smoking** (Yes vs No) | 1.32 [0.76–2.30] | 0.322 | | |
| **Hypertension** (Yes vs No) | 2.33 [1.77–3.08] | < 0.001 | 2.33 [1.77–3.07] | < 0.001* |
| **Diabetes** (Yes vs No) | 2.05 [1.13–3.70] | 0.017 | 1.96 [1.09–3.53] | 0.024* |
| **Anxiety** (Yes vs No) | 1.16 [0.81–1.67] | 0.413 | | |
| **Depression** (Yes vs No) | 1.11 [0.83–1.50] | 0.475 | | |

aOR: Adjusted odd ratio; Final model adjusted on age, gender, high blood pressure, diabetes and anxiety

[Ł]: p value of Wald test for binary variables or likelihood test for categorical variables with more than two modalities

*: Statistically significant

[1]: All the variance inflation factor (VIF) of the variables of the final model are ≤ 5.

anxiety are significantly correlated with IABPD [28]. A continual anxiety response raises BP, largely due to hormones and chemical reactions. Stress and anxiety do not only increase the workload on the cardiovascular system but also lead to sympathetic activation of the Renin-Angiotensin system [28].

In our study, increasing of age was associated with sIABPD ≥ 10 mmHg. This association has been demonstrated in previous studies [24, 29],as well as significant association between hypertension and sIABPD ≥ 10 mmHg [23–25, 30, 31] and also association between diabetes and sIABPD ≥ 10 mmHg [15, 29].

In sensitivity analysis, using IAD ≥ 15 mmHg as the cutoff for asymmetry, hypertension remained significantly associated with sIABPD. This result is consistent with the literature. Indeed, both UK and European guidelines recognize a systolic difference of 15 mmHg or more between the two arms as the threshold for additional cardiovascular risk [5, 32]. However, a recent study reports that a limit of 10 mm Hg may already leads to cardiovascular issues [9]. In contrast to the results from our main analysis, education level was found to be significantly associated with sIABPD ≥ 15 mmHg in sensitivity analysis. It is well known that educational inequality is one of the important factors that could increase the risk of CVD occurrence. Greater education tends to be associated with healthier behaviors, occupations with healthier working conditions, and better access to health care [33]. In addition, the increase in the cutoff point of the sIABPD value for sensitivity analyses may also explain the observed significance on this relation.

The main limitation of this study is the sequential measurement of BP, where beat-to-beat blood pressure variability explains in part the IABPD. We were unable to provide simultaneous measurement because of lack of specific machines. However, in clinical practice, and even more in low-income countries, BP is measured sequentially, so the clinical consequences

of our findings could be considered relevant. Another limitation of our study is the low repeatability of sIABPD. This could lead to variability in the prevalence of sIABPD in our study population. We also had no information about lipid levels and the use of lipid-lowering drugs in our population, these factors might be associated with sIABPD. Single point measurement for hypertension and hyperglycemia is also a limitation of our study as it could lead to an overestimation of their prevalence. The other limitation is the cross-sectional nature of our study so that we are not able to provide the prognostic value of systolic IABPD in our cohort. Long-term follow-up of our cohort will enable to provide this information.

In conclusion, this first study in an African-population, we report a high prevalence of sIABPD $\geq$ 10 mmHg in a rural population in Benin. Age, hypertension, and diabetes were significantly associated. Assessment of BP in both arms should become an essential component of clinical examination in general population, and cardiovascular risk assessment especially for individuals with hypertension or diabetes.

## Supporting information

**S1 File. Subdivision of Benin into departments and geographical location of the study area (Tanve area) in Benin.**
(DOCX)

**S2 File. Factors associated with systolic inter-arm blood pressure difference $\geq$ 15 mmHg, univariate and multivariate analysis, sensitivity analysis, TAHES study, Benin 2020.**
(DOCX)

**S3 File.**
(PDF)

**S4 File.**
(PDF)

## Acknowledgments

The authors thank Carine ATINDEHOU, Auriane ADJAHOUHOUE, Concheta TCHIBOZO, and Gilbert ASSOYIHIN for the quality of their daily work on TAHES. Thanks also to the participants in this survey, the regional health directorate of Zou, the Mayor of Agbangnizoun and his staff, the village of Tanvè chief ministry and the community health workers in Tanvè.

## Author Contributions

**Conceptualization:** Gwladys Nadia Gbaguidi, Audrey Kaboure, Yessito Corine Houehanou, Salimanou Ariyo Amidou, Dismand Stephan Houinato, Victor Aboyans, Philippe Lacroix.

**Data curation:** Gwladys Nadia Gbaguidi, Yessito Corine Houehanou, Salimanou Ariyo Amidou, Dismand Stephan Houinato, Victor Aboyans, Philippe Lacroix.

**Formal analysis:** Gwladys Nadia Gbaguidi, Audrey Kaboure, Yessito Corine Houehanou, Salimanou Ariyo Amidou, Dismand Stephan Houinato, Victor Aboyans, Philippe Lacroix.

**Funding acquisition:** Yessito Corine Houehanou, Salimanou Ariyo Amidou, Dismand Stephan Houinato, Victor Aboyans, Philippe Lacroix.

**Investigation:** Gwladys Nadia Gbaguidi, Yessito Corine Houehanou, Salimanou Ariyo Amidou, Dismand Stephan Houinato, Victor Aboyans, Philippe Lacroix.

**Methodology:** Gwladys Nadia Gbaguidi, Audrey Kaboure, Yessito Corine Houehanou, Salimanou Ariyo Amidou, Dismand Stephan Houinato, Victor Aboyans, Philippe Lacroix.

**Project administration:** Gwladys Nadia Gbaguidi, Yessito Corine Houehanou, Salimanou Ariyo Amidou, Dismand Stephan Houinato, Victor Aboyans, Philippe Lacroix.

**Resources:** Yessito Corine Houehanou, Salimanou Ariyo Amidou, Dismand Stephan Houinato, Victor Aboyans, Philippe Lacroix.

**Software:** Gwladys Nadia Gbaguidi, Audrey Kaboure, Yessito Corine Houehanou, Salimanou Ariyo Amidou, Dismand Stephan Houinato, Victor Aboyans, Philippe Lacroix.

**Supervision:** Gwladys Nadia Gbaguidi, Yessito Corine Houehanou, Salimanou Ariyo Amidou, Dismand Stephan Houinato, Victor Aboyans, Philippe Lacroix.

**Validation:** Gwladys Nadia Gbaguidi, Audrey Kaboure, Yessito Corine Houehanou, Salimanou Ariyo Amidou, Dismand Stephan Houinato, Victor Aboyans, Philippe Lacroix.

**Visualization:** Gwladys Nadia Gbaguidi, Audrey Kaboure, Yessito Corine Houehanou, Salimanou Ariyo Amidou, Dismand Stephan Houinato, Victor Aboyans, Philippe Lacroix.

**Writing – original draft:** Gwladys Nadia Gbaguidi, Audrey Kaboure, Yessito Corine Houehanou, Salimanou Ariyo Amidou, Dismand Stephan Houinato, Victor Aboyans, Philippe Lacroix.

**Writing – review & editing:** Gwladys Nadia Gbaguidi, Audrey Kaboure, Yessito Corine Houehanou, Salimanou Ariyo Amidou, Dismand Stephan Houinato, Victor Aboyans, Philippe Lacroix.

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
