## [Decision Letter · Decision Letter 0]

5 Oct 2021

PONE-D-21-28514Inter-Arm Difference in Systolic Blood Pressure: Prevalence and Associated Factors in an African PopulationPLOS ONE

Dear Dr. GBAGUIDI,

Thank you for submitting your manuscript to PLOS ONE. After careful consideration, we feel that it has merit but does not fully meet PLOS ONE’s publication criteria as it currently stands. Therefore, we invite you to submit a revised version of the manuscript that addresses the points raised during the review process.

Both Reviewers raised concern on the data analyses of this manuscript, please consider carefully the suggestions of the Reviewers during the revision. Especially, please provide more information on the risk factors of atherosclerotic diseases, such as cardiovascular disease history, serum lipid level, dyslipidemia or use of statins, etc., and consider them in the multivariate analysis as appropriate.

We look forward to receiving your revised manuscript.

Kind regards,

Yan Li, MD, PhD

Academic Editor

PLOS ONE

Journal Requirements:

Reviewers' comments:

Reviewer's Responses to Questions

**Comments to the Author**

1. Is the manuscript technically sound, and do the data support the conclusions?

Reviewer #1: No

Reviewer #2: Yes

2. Has the statistical analysis been performed appropriately and rigorously? 

Reviewer #1: No

Reviewer #2: Yes

3. Have the authors made all data underlying the findings in their manuscript fully available?

Reviewer #1: No

Reviewer #2: Yes

4. Is the manuscript presented in an intelligible fashion and written in standard English?

Reviewer #1: Yes

Reviewer #2: Yes

5. Review Comments to the Author

Reviewer #1: In this manuscript, the authors investigated the prevalence and associated factors of inter-arm difference in systolic blood pressure in an African population. A total of 1,505 participants (women 59%) were included. They found that the prevalence of sIABPD ≥ 10 mmHg was 19% (95% CI: 17–21%). Moreover, the probability of sIABPD ≥ 10 mmHg increased significantly with age, hypertension and diabetes.

1. Prevalence of sIABPD rises in relation to underlying cardiovascular comorbidities of the population studied. Therefore, the prevalence of peripheral arterial disease, CAD, or cerebrovascular disease should be recorded in your manuscript.

2. BP is a variable hemodynamic phenomenon that constantly fluctuates over time, making sequential measurements difficult to compare, therefore, to prevent overestimation and observer bias, sIABPD should be assessed simultaneously at both arms.

3. Lipid related indicators should be included and analyzed as risk factors, such as triacylglycerol, total cholesterol and LDL-C. Moreover, the use of anti-hypertensive drugs, anti-diabetic agents，statins should be recorded and adjusted in your Multivariate analysis.

4. The repeatability of sIABPD is poor, which should be mentioned in the limitation.

5. The results are best to be illustrated in paragraphs:

1）study population

2）Prevalence of sIABP difference

3）Risk factors for sIABP difference

Moreover, the content of the results should be stated more specifically.

6. In Statistical analysis, you mentioned that‘The interactions between variables included in multivariable model were tested as well as linearity of quantitative variables’. While, no related results cannot be found in your manuscript.

7. Your manuscript only excluded pregnant woman, while participants lacking of systolic BP or diastolic BP in both arms or lacking of other risk factors should also be excluded. Moreover, the accurate number or percentage of missing values should be illustrated.

8. In the results, you mentioned that ‘An increase in the prevalence of sIABPD by age group was observed in both men and women (Figure 3)’. While the P value and P-trend value in Figure 3 should be calculated.

9. Why the Multivariate analysis included both Initial model and Final model?

Reviewer #2: This study explored the prevalence and risk factors of inter-arm blood pressure difference in Tanve village. Despite the great efforts of the research team, I think this article cannot yet conclude that IABPD can assess cardiovascular risk.

First, the medical history of subjects was missing in the multiple regression model, especially the history of cardiovascular disease, such as coronary artery disease, stroke or peripheral arteria disease.

Second, the definition of diabetes in this study, which was defined by fasting blood glucose value ≥ 7 mmol/L or currently taking diabetes medication, did not meet the existing diagnostic guidelines.

In addition, I think that the subjects' lipid levels and their use of lipid-lowering drugs will also be associated with IABPD, which was unfortunately not mentioned in the article.

Finally, what is the relationship between different office blood pressure levels and IABPD in this study? Consider including stratified analysis of office blood pressure or as a continuous variable into multiple regression model analysis.

6. PLOS authors have the option to publish the peer review history of their article (what does this mean?). If published, this will include your full peer review and any attached files.

Reviewer #1: No

Reviewer #2: No

---

## [Author Response · Author response to Decision Letter 0]

1 Dec 2021

Dear editor,

We thank you for giving us the opportunity to revise and resubmit our manuscript. We also thank the reviewers for the helpful comments aiming at improving the article. We propose here a revised version accounting for the comments and suggestions made by the reviewers. We remain available for further request.

Best regards

PONE-D-21-28514

Inter-Arm Difference in Systolic Blood Pressure: Prevalence and Associated Factors in an African Population

PLOS ONE

Comments to the Author

1. Is the manuscript technically sound, and do the data support the conclusions?

Reviewer #1: No

Reviewer #2: Yes

2. Has the statistical analysis been performed appropriately and rigorously?

Reviewer #1: No

Reviewer #2: Yes

3. Have the authors made all data underlying the findings in their manuscript fully available?

Reviewer #1: No

Reviewer #2: Yes

4. Is the manuscript presented in an intelligible fashion and written in standard English?

Reviewer #1: Yes

Reviewer #2: Yes

Reviewer reports:

Reviewer 1: 

1. Prevalence of sIABPD rises in relation to underlying cardiovascular comorbidities of the population studied. Therefore, the prevalence of peripheral arterial disease, CAD, or cerebrovascular disease should be recorded in your manuscript. 

We thank the reviewer for these comments. We cannot provide information on peripheral arterial disease because the ankle-brachial index was not measured in 2020. Also, the level of diagnosis of coronary artery disease and cerebrovascular disease is very low due to under-medicalization. However, we provided in table 1 information on participants’ history of cardiovascular and neurological diseases.

2. BP is a variable hemodynamic phenomenon that constantly fluctuates over time, making sequential measurements difficult to compare, therefore, to prevent overestimation and observer bias, sIABPD should be assessed simultaneously at both arms. 

Thank you for this comment. We agree with the beat-to-beat variability of BP, but devices measuring simultaneously both arms are not available in practice in Africa, so the clinical interest is questionable. Also, most studies on IABPD have been performed with sequential measurement. This point has been highlighted in the discussion.

3. Lipid related indicators should be included and analyzed as risk factors, such as triacylglycerol, total cholesterol and LDL-C. Moreover, the use of anti-hypertensive drugs, anti-diabetic agents，statins should be recorded and adjusted in your Multivariate analysis. 

We have presented all the available variables in Tables 1. However, regarding to the insertion of preventive therapies variable in the multivariate model, this lead a methodological concern. Indeed, there was a strong collinearity between the use of a treatment (e.g. antihypertensive therapy) and presence of factor such as Hypertension, or high blood pressure in the same model. Because the level of use of medication very low in this low income community with low medical resource, we preferred to use the risk factors, rather than the treatment, in the models. The proportion of hypertensive participants on treatment has been reported in the results section.

4. The repeatability of sIABPD is poor, which should be mentioned in the limitation.

We thank reviewer for this comment. We have considered this point in the limitation in discussion section.

5. The results are best to be illustrated in paragraphs: 

We have considered this point in the result section

1）Study population

2）Prevalence of sIABP difference

3）Risk factors for sIABP difference

Moreover, the content of the results should be stated more specifically.

We have taken this comment in consideration in the results section.

6. In Statistical analysis, you mentioned that‘The interactions between variables included in multivariable model were tested as well as linearity of quantitative variables’. While, no related results cannot be found in your manuscript. 

These analyses are multiple and would dramatically lengthen the results chapter. This kind of analysis is rarely presented in manuscripts. We can send to the editorial team the statistical reports if requested. They have been performed.

7. Your manuscript only excluded pregnant woman, while participants lacking of systolic BP or diastolic BP in both arms or lacking of other risk factors should also be excluded. Moreover, the accurate number or percentage of missing values should be illustrated. 

All participants had bilateral measurements and data on risk factors collected. 

8. In the results, you mentioned that ‘An increase in the prevalence of sIABPD by age group was observed in both men and women (Figure 3)’. While the P value and P-trend value in Figure 3 should be calculated.

We calculated the p value of the association between sex and sIABPD in each age group.

9. Why the Multivariate analysis included both Initial model and Final model? 

Any multivariate analysis with stepwise approach has an initial and then final model. We considered fair to present both so to bring more information. We think that the reader would better understand where we have started and where we have ended by showing both.

 

Reviewer #2: 

1. First, the medical history of subjects was missing in the multiple regression model, especially the history of cardiovascular disease, such as coronary artery disease, stroke or peripheral arteria disease. 

We thank the reviewer for this comment. As mentioned earlier, we cannot provide information on peripheral arterial disease because the ankle-brachial index was not measured in 2020. Also, the level of diagnosis of coronary artery disease and cerebrovascular disease is very low due to under-medicalization. However, we provided information on cardiovascular and neurological conditions reported by the participants.

2. Second, the definition of diabetes in this study, which was defined by fasting blood glucose value ≥ 7 mmol/L or currently taking diabetes medication, did not meet the existing diagnostic guidelines. 

The definition used for diabetes is those recommended by World Health Organization for STEPS surveys. In addition, the cohort is dynamic, so there are participants who have never been evaluated in previous years. Therefore, we cannot use only the data from previous years to define the history. As we performed single point measurement of blood pressure and glycemia, we have added within the limit section (p 10) the following: “Single point measurement for hypertension and hyperglycemia is also a limitation of our study as it could lead to an overestimation of their prevalence”.

3. In addition, I think that the subjects' lipid levels and their use of lipid-lowering drugs will also be associated with IABPD, which was unfortunately not mentioned in the article. 

We agree with the reviewers' comment. However, we do not have data on the participants’ lipid levels or the use of lipid-lowering drugs. We have added this point in the limitation in discussion section.

4. Finally, what is the relationship between different office blood pressure levels and IABPD in this study? Consider including stratified analysis of office blood pressure or as a continuous variable into multiple regression model analysis. 

We have not done office BP measurements. Therefore, we cannot determine what the relationship is between the sIABPD and the office BP measurements found in a practice. However, in this new version of the manuscript, we have presented a figure that shows the relationship between the sIABPD and the conventional BP classification groups.

---

## [Decision Letter · Decision Letter 1]

9 Feb 2022

PONE-D-21-28514R1Inter-Arm Difference in Systolic Blood Pressure: Prevalence and Associated Factors in an African PopulationPLOS ONE

Dear Dr. GBAGUIDI,

Thank you for submitting your manuscript to PLOS ONE. After careful consideration, we feel that it has merit but does not fully meet PLOS ONE’s publication criteria as it currently stands. Therefore, we invite you to submit a revised version of the manuscript that addresses the points raised during the review process.

Please consider the additional comments of Reviewer 1 in your revision. In addition, could you try to use the inter-arm SBP difference as a continuous variable to investigate the determinants, and perform sensitivity analyses using 15 mm Hg as a cut-off of the large inter-arm BP difference?

We look forward to receiving your revised manuscript.

Kind regards,

Yan Li, MD, PhD

Academic Editor

PLOS ONE

Journal Requirements:

Reviewers' comments:

Reviewer's Responses to Questions

**Comments to the Author**

1. If the authors have adequately addressed your comments raised in a previous round of review and you feel that this manuscript is now acceptable for publication, you may indicate that here to bypass the “Comments to the Author” section, enter your conflict of interest statement in the “Confidential to Editor” section, and submit your "Accept" recommendation.

Reviewer #1: (No Response)

Reviewer #2: All comments have been addressed

2. Is the manuscript technically sound, and do the data support the conclusions?

Reviewer #1: Partly

Reviewer #2: Yes

3. Has the statistical analysis been performed appropriately and rigorously? 

Reviewer #1: No

Reviewer #2: Yes

4. Have the authors made all data underlying the findings in their manuscript fully available?

Reviewer #1: Yes

Reviewer #2: Yes

5. Is the manuscript presented in an intelligible fashion and written in standard English?

Reviewer #1: Yes

Reviewer #2: Yes

6. Review Comments to the Author

Reviewer #1: Thanks for your modification, while there exist minor questions as following.

1. Table 2 and 3 in Result should be capitalized.

2. The P value of female in Table 3 is missing.

3. You need not to show me the specific process of collinearity, while you need to tell me the final results about the collinearity. Because the backward stepwise method maybe not solid when the independent variables are highly correlated. Furthermore, I wonder why you chose backward instead of stepwise logistic regression?

4. In your manuscript, I suppose that the initial model refers to the result of general multiple logistic regression, while the final model refers to the result of backward stepwise selection with the variables which P value less than 0.2 in Initial model. Is my understanding correct? Dose there exist any reference for similar analyses?

Reviewer #2: (No Response)

7. PLOS authors have the option to publish the peer review history of their article (what does this mean?). If published, this will include your full peer review and any attached files.

Reviewer #1: No

Reviewer #2: No

---

## [Author Response · Author response to Decision Letter 1]

4 Apr 2022

Response to Reviewers

Dear editor,

Thank you to giving again the opportunity to review and resubmit our manuscript. We also thank the reviewer for his/her additional comments. A revised version taking into account the reviewer’ comments and journal requirements is provided here. 

Best regards

PONE-D-21-28514R1

Inter-Arm Difference in Systolic Blood Pressure: Prevalence and Associated Factors in an African Population

PLOS ONE

Journal requirements 

Please consider the additional comments of Reviewer 1 in your revision. In addition, could you try to use the inter-arm SBP difference as a continuous variable to investigate the determinants, and perform sensitivity analyses using 15 mm Hg as a cut-off of the large inter-arm BP difference?

We thank the editor for his/her comments. We have responded to the questions of reviewer 1 and conducted a sensitivity analysis using 15 mm Hg as a cut-off of the large inter-arm BP difference as requested.

Regarding suggestion related to the use of inter-arm SBP difference as a continuous variable to investigate the determinants, we observed that all assumptions and conditions for linear regression (e.g., the normality of error distribution and homoscedasticity) were not satisfied. Therefore, we can’t run linear regression as our predictions won’t be accurate. 

Reference list have been checked. No retracted articles have been cited in our work.

Reviewer #1: Thanks for your modification, while there exist minor questions as following.

1. Table 2 and 3 in Result should be capitalized.

We thank reviewer for this comment. We have taken this comment in consideration in the results section.

2. The P value of female in Table 3 is missing.

This information is now provided in the Table 3.

3. You need not to show me the specific process of collinearity, while you need to tell me the final results about the collinearity. Because the backward stepwise method maybe not solid when the independent variables are highly correlated. Furthermore, I wonder why you chose backward instead of stepwise logistic regression?

The stepwise approches includes backward and forward selection. 

Backward selection - starting with the full model has the advantage of considering the effects of all variables simultaneously. In contrast with the reviewer statement, this selection approach is especially important when variables in a model are correlated which each other as backward stepwise may be forced to keep them all in the model unlike forward selection where none of them might be entered. In addition, R and other software automatically calculate the best thresholds required to identify the best model (model with the highest likelihood, the one with the lowest Akaike information criterion). 

4. In your manuscript, I suppose that the initial model refers to the result of general multiple logistic regression, while the final model refers to the result of backward stepwise selection with the variables which P value less than 0.2 in Initial model. Is my understanding correct? Dose there exist any reference for similar analyses?

The initial model includes the variables which p-value less than 0.2 in univariables analysis. Except, age and sex that we imposed in the final model, the variables in this model are those obtained at the end of the backward stepwise selection. Compared to initial model, the final model was those with the highest likelihood, the one with the lowest Akaike information criterion (Akaike value of initial model is 1434 and which for final model is 1426). Then, in this final model, only the variables which have a p-value less than 0.05 were considered as statistically associated with our dependent variable. 

More detail about the backward stepwise selection method are describe in this manuscript “Sanchez-Pinto LN, Venable LR, Fahrenbach J, Churpek MM. Comparison of variable selection methods for clinical predictive modeling. Int J Med Inform. 2018 Aug;116:10-17. doi: 10.1016/j.ijmedinf.2018.05.006. Epub 2018 May 21. PMID: 29887230; PMCID: PMC6003624. »

---

## [Decision Letter · Decision Letter 2]

16 May 2022

PONE-D-21-28514R2Inter-Arm Difference in Systolic Blood Pressure: Prevalence and Associated Factors in an African PopulationPLOS ONE

Dear Dr. GBAGUIDI,

Thank you for submitting your manuscript to PLOS ONE. After careful consideration, we feel that it has merit but does not fully meet PLOS ONE’s publication criteria as it currently stands. Therefore, we invite you to submit a revised version of the manuscript that addresses the points raised during the review process.

Reviewer 1 still has some suggestions about the revision. Please carefully consider.

We look forward to receiving your revised manuscript.

Kind regards,

Yan Li, MD, PhD

Academic Editor

PLOS ONE

Journal Requirements:

Reviewers' comments:

Reviewer's Responses to Questions

**Comments to the Author**

1. If the authors have adequately addressed your comments raised in a previous round of review and you feel that this manuscript is now acceptable for publication, you may indicate that here to bypass the “Comments to the Author” section, enter your conflict of interest statement in the “Confidential to Editor” section, and submit your "Accept" recommendation.

Reviewer #1: (No Response)

Reviewer #2: All comments have been addressed

2. Is the manuscript technically sound, and do the data support the conclusions?

Reviewer #1: Yes

Reviewer #2: Yes

3. Has the statistical analysis been performed appropriately and rigorously? 

Reviewer #1: Yes

Reviewer #2: Yes

4. Have the authors made all data underlying the findings in their manuscript fully available?

Reviewer #1: Yes

Reviewer #2: Yes

5. Is the manuscript presented in an intelligible fashion and written in standard English?

Reviewer #1: Yes

Reviewer #2: Yes

6. Review Comments to the Author

Reviewer #1: 1、 The actual contents of Supplement 1 and 2 are inconsistent with the supposed contents.

2、 The main approaches for stepwise regression are: Forward selection, Backward elimination and Bidirectional elimination (a combination of the above, testing at each step for variables to be included or excluded.), instead of only backward and forward selection. Backward elimination is especially important in case of collinearity, while whether the multivariable model have multicollinearity or not? Please give the Tolerance value and/or VIF value in the results.

3、 Please add to the Discussion of your interpretation for the consistent and inconsistent results between sensitivity analyses and the main analyses.

Reviewer #2: (No Response)

7. PLOS authors have the option to publish the peer review history of their article (what does this mean?). If published, this will include your full peer review and any attached files.

Reviewer #1: No

Reviewer #2: No

---

## [Author Response · Author response to Decision Letter 2]

13 Jun 2022

Response to Reviewers

Dear editor,

Thank you to giving again the opportunity to review and resubmit our manuscript. We also thank the reviewer for his/her additional comments. A new revised version taking into account the reviewer’ comments is provided here. 

Best regards

PONE-D-21-28514R1

Inter-Arm Difference in Systolic Blood Pressure: Prevalence and Associated Factors in an African Population

PLOS ONE

Reviewer #1:

The actual contents of Supplement 1 and 2 are inconsistent with the supposed contents.

We thank reviewer for this comment. Supplement 1 shows the location of the study area on the Benin map . Supplement 2 presents the results from the sensitivity analysis.

The main approaches for stepwise regression are: Forward selection, Backward elimination and Bidirectional elimination (a combination of the above, testing at each step for variables to be included or excluded.), instead of only backward and forward selection. Backward elimination is especially important in case of collinearity, while whether the multivariable model have multicollinearity or not? Please give the Tolerance value and/or VIF value in the results.

Please find below the VIF values of each variable in this model. All VIF were lower than 5 suggesting that there no multicollinearity issues in our final model. As suggested by the reviewer, we added this information in results section in the table 3.

 Age Gender Hypertension Diabetes 

VIF 1.11 1.00 1.10 1.00

Please add to the Discussion of your interpretation for the consistent and inconsistent results between sensitivity analyses and the main analyses.

We thank reviewer for this comment. We have taken this comment in consideration in the discussion section. (See Page 10)

---

## [Decision Letter · Decision Letter 3]

25 Jul 2022

Inter-Arm Difference in Systolic Blood Pressure: Prevalence and Associated Factors in an African Population

PONE-D-21-28514R3

Dear Dr. GBAGUIDI,

We’re pleased to inform you that your manuscript has been judged scientifically suitable for publication and will be formally accepted for publication once it meets all outstanding technical requirements.

Kind regards,

Yan Li, MD, PhD

Academic Editor

PLOS ONE

Additional Editor Comments (optional):

Reviewers' comments:

Reviewer's Responses to Questions

**Comments to the Author**

1. If the authors have adequately addressed your comments raised in a previous round of review and you feel that this manuscript is now acceptable for publication, you may indicate that here to bypass the “Comments to the Author” section, enter your conflict of interest statement in the “Confidential to Editor” section, and submit your "Accept" recommendation.

Reviewer #1: All comments have been addressed

Reviewer #2: (No Response)

2. Is the manuscript technically sound, and do the data support the conclusions?

Reviewer #1: Yes

Reviewer #2: Yes

3. Has the statistical analysis been performed appropriately and rigorously? 

Reviewer #1: Yes

Reviewer #2: Yes

4. Have the authors made all data underlying the findings in their manuscript fully available?

Reviewer #1: Yes

Reviewer #2: Yes

5. Is the manuscript presented in an intelligible fashion and written in standard English?

Reviewer #1: Yes

Reviewer #2: Yes

6. Review Comments to the Author

Reviewer #1: (No Response)

Reviewer #2: (No Response)

7. PLOS authors have the option to publish the peer review history of their article (what does this mean?). If published, this will include your full peer review and any attached files.

Reviewer #1: No

Reviewer #2: No

---

## [Editor Report · Acceptance letter]

22 Aug 2022

PONE-D-21-28514R3 

Inter-Arm Difference in Systolic Blood Pressure: Prevalence and Associated Factors in an African Population. 

Dear Dr. Gbaguidi:

I'm pleased to inform you that your manuscript has been deemed suitable for publication in PLOS ONE. Congratulations! Your manuscript is now with our production department. 

Kind regards, 

on behalf of

Professor Yan Li 

Academic Editor

PLOS ONE